# Exploring the Molecular Mechanism and Role of Glutathione S-Transferase P in Prostate Cancer

**DOI:** 10.3390/biomedicines13051051

**Published:** 2025-04-26

**Authors:** Shan Huang, Hang Yin

**Affiliations:** 1Department of Urology, Beijing Chao-Yang Hospital, Capital Medical University, Beijing 100020, China; 18661656191@163.com; 2Institute of Urology, Beijing Chao-Yang Hospital, Capital Medical University, Beijing 100020, China

**Keywords:** Glutathione S-transferase P, prostate cancer, Mendelian randomization, causality, machine learning, single-cell analysis

## Abstract

**Aims:** To investigate the effect of Glutathione metabolism in prostate cancer pathogenesis. **Background**: There is growing evidence that Glutathione metabolism plays an important role in prostate cancer, with genes encoding key enzymes in this pathway potentially serving as diagnostic or prognostic biomarkers. **Objective**: To explore whether there is a causal relationship between key enzymes in the Glutathione metabolism and prostate cancer, and to further investigate the molecular mechanisms and roles of the genes encoding their proteins in relation to prostate cancer. **Method**: Transcriptomic datasets from the Gene Expression Omnibus (GEO) database were analyzed to identify differentially expressed genes (DEGs) and enriched pathways in prostate cancer versus normal tissues. Two-sample bidirectional Mendelian randomization (MR) was employed to assess causal relationships between Glutathione metabolic enzymes (exposure) and prostate cancer risk (outcome). Immune infiltration analysis and LASSO regression were performed to construct a diagnostic model. Single-cell RNA sequencing (scRNA-seq) data were utilized to elucidate cell-type-specific expression patterns and functional associations of target genes. **Result**: The results of two-sample bidirectional MR showed that Glutathione S-transferase P (GSTP) in Glutathione metabolism could reduce the risk of prostate cancer. The Glutathione S-transferase Pi-1 (GSTP1) gene was lowly expressed in prostate cancer and was able to diagnose prostate cancer more accurately. Single-cell analysis showed that the high expression of GSTP1 in prostate cancer epithelial cells was closely associated with the upregulation of the P53 pathway and apoptosis. **Conclusions**: Our study reveals that GSTP in Glutathione metabolism reduces the risk of prostate cancer and further analyzes the genetic association and mechanism of action between GSTP1 and prostate cancer.

## 1. Introduction

Prostate cancer is the most common tumor in European and American men [1,2]. Epidemiological projections estimate 299,010 new diagnoses and 35,250 prostate-cancer-related deaths in the United States during 2024 [2]. Established risk factors include advanced age, African ancestry, and familial predisposition to the disease [3]. Tumors have common metabolic features, but some studies have shown that prostate cancer itself has unique metabolic features: abnormalities in amino acid metabolism (particularly glutamate metabolism), abnormalities in cholesterol metabolism, etc. [4]. These metabolic insights are informing novel therapeutic strategies and monitoring approaches for clinical management.

Glutathione (GSH), an intracellular antioxidant, is the most abundant non-protein thiol at millimolar concentrations in mammals [5]. Production of reactive oxygen species in most tumor cells can lead to increased GSH levels and the expression of the antioxidant process [5]. And as an important antioxidant regulator, NRF2, which regulates the expression of several enzymes in the Glutathione metabolism, including Glutathione reductase, Glutathione peroxidase, and Glutathione S-transferase (GST) [6,7]. GST can prevent cancer progression [5]. Its polymorphisms are associated with susceptibility to a variety of cancers [8,9]. A study has shown that GSTP1 polymorphisms behave differently in different tumors, with the GSTP1a allele being significantly lower in prostate cancer while the GSTP1b allele is significantly higher in bladder and testicular cancer [10]. However, the mechanistic link between GSTP1 polymorphisms and prostate carcinogenesis remains controversial in contemporary research.

This study aimed to investigate the association between key Glutathione metabolic enzymes and prostate cancer. MR analysis, an epidemiological method grounded in Mendelian inheritance principles, employs genetic variants as instrumental variables to infer causal relationships between exposures (e.g., enzyme activity) and outcomes (e.g., prostate cancer risk), thereby minimizing confounding and reverse causality. MR analysis was utilized to elucidate causal links between Glutathione metabolic enzymes and prostate cancer. Concurrently, machine learning approaches were applied to evaluate the diagnostic utility of key enzyme-related genes and construct predictive models. Single-cell resolution analysis further delineated cell-type-specific expression patterns of these genes. This integrative approach aims to advance the identification of novel therapeutic targets for prostate cancer. The study design is summarized in Appendix A.

## 2. Materials and Methods

### 2.1. Dataset Download from GEO Database and Data Preprocessing

In this study, four prostate cancer datasets were retrieved from the GEO database. The GSE62872 dataset served as a validation set to confirm our findings, while the remaining three datasets were utilized to identify DEGs in prostate cancer. Following batch correction and normalization of the three datasets, the merged data were subjected to differential expression analysis to obtain DEGs. These genes were subsequently analyzed via Gene Ontology (GO) and Kyoto Encyclopedia of Genes and Genomes (KEGG) enrichment analyses to identify enriched pathways. Additionally, the GSE193337 dataset was downloaded for single-cell analysis. Details of the downloaded datasets are summarized in Table 1, with comprehensive clinical information for each dataset provided in Appendix A.

### 2.2. Download of Data from the GWAS Public Database

Enriched pathway datasets were obtained from the GWAS Catalog (https://gwas.mrcieu.ac.uk/ (accessed on 5 August 2024)) for use as exposure variables in MR analysis. A prostate cancer genome-wide association study (GWAS) dataset (ieu-b-85) was acquired as the outcome variable. All data originated from European ancestry populations. Summary statistics utilized in these MR analyses were derived from published studies where ethical approval and participant consent had been secured in the original research. The downloaded datasets are cataloged in Table 1.

### 2.3. Instrumental Variables (IVs)

The MR analysis used genetic variants as IVs to examine whether there was a causal relationship between exposure factors and outcome factors. To avoid bias, we took the following measures to treat the exposure factors strictly: (1) We set the significance level of the correlation analysis at *p* < 5 × 10^−6^; (2) To minimize linkage disequilibrium (LD), we set a clustering window of 10,000 kilobases and an LD threshold of r^2^ < 0.001; (3) We calculated the F-statistics of IVs, excluded relevant IVs with F ≤ 10, and retained the remaining IVs for further analysis to avoid the weak instrumental bias caused by variable effects. Two-sample bidirectional MR analyses of exposure and outcome factors were then performed.

### 2.4. MR Analysis Methods

The MR analyses employed five approaches: inverse variance weighting (IVW), MR-Egger regression, weighted median (WM), simple mode, and weighted mode. The IVW assumes that the instrumental variables are free of horizontal multivariate, which has the advantage of high test validity but may lead to bias if multivariate is present. MR-Egger regression assumes that the instrumental variables satisfy the InSIDE assumptions and that SNP exposure associations are sufficiently precise; however, MR-Egger is sensitive to outliers and violations of the InSIDE assumptions, and the test is inefficient [11]. Weighted median assumes that at least 50% of the instrumental variables are valid, and the method is robust to outliers, and sensitive to additions or removals of genetic variants. To make the MR analysis more complete, we used both simple and weighted models.

Sensitivity analysis: Heterogeneity was assessed using Cochran’s Q test within the IVW framework, where *p* > 0.05 indicated no significant heterogeneity. Horizontal pleiotropy was evaluated via funnel plots and MR-Egger intercept analysis, with the latter quantifying the magnitude of pleiotropic effects.

### 2.5. Construct Drug-Gene Regulatory Network and ceRNA Regulatory Network of Pathway Genes

Drug–pathway gene interactions were identified using the DGIdb database (https://dgidb.org/ (accessed on 5 August 2024)), and the drug-gene regulatory network was constructed via Cytoscape (version 3.9.1). miRNA-target relationships for pathway genes were screened across four databases: miRanda, miRDB, miRWalk, and TargetScan. Subsequently, lncRNA–miRNA binding pairs were analyzed using the spongeScan database to establish a competing endogenous RNA (ceRNA) regulatory network.

### 2.6. Analysis of Immune Infiltration of Pathway Genes

The CIBERSORT package was applied to analyze the relative expression of pathway genes in different immune cells and to further visualize the interactions between immune cells and the correlation between pathway genes and immune cells.

### 2.7. Diagnostic Ability, Expression Level, and Validation of Pathway Genes

To evaluate the diagnostic utility of pathway genes, receiver operating characteristic (ROC) analysis was performed using the pROC package (v1.18.5), with area under the curve (AUC) values calculated to quantify classification accuracy between control and case cohorts. Differential expression of pathway genes was assessed in the training cohort via the ggpubr package (v0.6.0), with results independently validated in the test cohort. Immunohistochemistry (IHC) further corroborated gene expression patterns in matched normal and tumor tissues.

### 2.8. Construction of Prostate Cancer Prediction Model Based on Machine Learning and Pathway Genes

The pathway genes were screened using LASSO regression, and the obtained hub genes were used as the final genes for constructing the prostate cancer prediction model. The results were visualized by the “pROC” package.

### 2.9. Processing of Single-Cell RNA-Seq Data

To delineate the cellular expression landscape of pathway genes in prostate cancer, two samples (GSM5793824 and GSM5793828) from the GSE193337 dataset were processed using Seurat (v4.4.0). First, raw data underwent stringent quality filtering: cells with <200 or >10,000 unique molecular identifiers (UMIs), mitochondrial gene content > 20%, or ribosomal RNA content > 20% were excluded. Normalization was performed using the LogNormalize method. The “FindVariableFeatures” function was then applied to obtain 2000 highly variable genes, which were used for subsequent PCA downscaling analysis. Second, after normalizing all genes combined with jackstraw and elbow plots, we determined the principal component (PC) to be 15 and further selected a resolution of 1.2 for PCA downscaling. Third, the UMAP nonlinear dimensionality reduction algorithm was applied to display cell clusters. Fourth, seven cell types (T cells; epithelial cells; endothelial cells; NK cells; monocytes; tissue stem cells and smooth muscle cells) were assigned to cell clusters based on the genetic markers by “SingleR”. The markers for each cell cluster are listed in Appendix A and are also visualized as heatmaps of pathway gene expression in different cell types. Fifth, the high and low expression of GSTP1 in epithelial cells were individually characterized. And the pathway genes were enriched and scored by calling the msigdbr package using the “irGSEA” package in “UCell” (v 2.6.2).

### 2.10. Statistical Analysis

This study was analyzed using R software (v 4.3.0). The Wilcoxon rank sum test was used to analyze the differential expression levels of pathway genes. Spearman correlation analysis was used to explore the correlation between pathway genes and immune cells. A *p*-value of <0.05 was considered statistically significant.

## 3. Results

### 3.1. Data Merging and Gene Differential Expression Analysis

We have corrected and completed the multi-chip merging of the three data, GSE46602, GSE55945, and GSE69223. Using R software (version 4.3.0), we performed principal component analysis (PCA) on the three data before and after data merging. The results demonstrated successful batch effect removal in the integrated dataset, with minimal residual systematic variation between the three expression matrices (Figure 1A). The integrated expression matrix was analyzed for differences using the limma package in R software, and significant differential genes were obtained based on |logFC| > 1 and adj.P.Val < 0.05 (Appendix A). Filter the top 50 significantly upregulated and downregulated differentially expressed genes (DEGs) and visualize them using the pheatmap package to generate heatmaps and volcano plots. (Figure 1B,C).

### 3.2. GO and KEGG Analysis of Differentially Expressed Genes

We visualized GO analysis and KEGG analysis using R software (Figure 2A,B). GO analysis was used to classify and annotate the functions of significantly differentially expressed genes, which were mainly classified into the following three categories: cellular components (CC), molecular functions (MF), and biological processes (BP). The classification leads to a better understanding of the functions of genes and the biological processes in which they are involved. In this study, the differentially expressed genes were found to be significantly associated with the extracellular matrix, epithelial–mesenchymal transition, and Glutathione transferase activity (Appendix A). KEGG analysis correlates significantly differentially expressed genes with known metabolic pathways to better understand the role of differentially expressed genes in organisms. This study showed that the differentially expressed genes were significantly associated with cytochrome P450, Glutathione metabolism, and prostate cancer (Appendix A).

### 3.3. MR Analysis

We selected the Glutathione metabolism for subsequent MR analysis based on the KEGG results and visualized the genes enriched in this pathway to obtain the volcano diagram (Figure 3A). Fourteen enzymes in the Glutathione metabolism were screened from the GWAS database as exposure factors, prostate cancer data were screened as the outcome factor, the TwoSampleMR package was applied to perform a two-sample bidirectional MR analysis, and the results were dominated by the IVW method. The IVW method provided two positive results: Glutathione S-transferase P (GSTP) was significantly causal in reducing the risk of prostate cancer (OR = 0.962, 95% CI 0.926–0.998, *p* = 0.038), Glutathione S-transferase omega-1 (GSTO1) also had a significant causal effect on prostate cancer risk reduction (OR = 0.977, 95% CI 0.959–0.995, *p* = 0.013), neither of which detected heterogeneity, with *p* values of 0.280 and 0.155 for Cochran’s Q, respectively. Reverse Mendelian randomization yielded no positive results. The sensitivity analysis can be found in Appendix A. In the test of multiplicity between GSTP and prostate cancer, the intercept was −0.006, with a *p*-value of 0.465, which was not statistically significant, and in the test of multiplicity between GSTO1 and prostate cancer, the intercept was −0.002, with a *p*-value of 0.749, which was not statistically significant. The results of the leave-one-out method showed that none of the SNPs had a significant impact on causality and had good robustness (Figure 3B,F). The forest plot of the MR analysis results indicated that IVW was statistically significant (Figure 3C,G). The funnel plot showed that the distribution of SNPs tended to be balanced (Figure 3D,H). The scatter plot indicated that the exposure factors were negatively correlated with the outcome factors (Figure 3E,I). MR-PRESSO analyses showed no horizontal multiplicity in causality for prostate cancer for both GSTP and GSTO1 (global test, *p* = 0.307 and *p* = 0.347), and neither had outliers. However, the differentially expressed genes enriched in the Glutathione metabolism did not have GSTO1, so we subsequently focused on GSTP in the analysis. In summary, Glutathione metabolism is associated with prostate cancer, and GSTP in the metabolic pathway of prostate cancer may reduce its risk.

### 3.4. Construct Drug-Gene Regulatory Network and ceRNA Regulatory Network

The DGIdb database (https://dgidb.genome.wustl.edu/ (accessed on 5 August 2024)) is a publicly available database of drug–gene interactions through which we have found a variety of drugs capable of regulating pathway genes (Table 2). The results were visualized by Cytoscape software (version 3.9.1) to obtain a diagram of the drug-gene regulatory network (Figure 4A); lncRNA binds miRNAs competitively, which in turn affects the miRNA-regulated target gene mRNAs. This extensive interaction between genes mediated by miRNAs is called the ceRNA regulatory network. Regarding MiRanda, miRDB, miRWalk, and TargetScan, all four databases were able to predict the miRNAs that regulate the target genes, and in this study, we picked out the miRNAs that can bind to pathway genes. Then, the spongeScan database was used to screen the lncRNAs that could bind to the above-mentioned miRNAs, and the ceRNA regulatory network was visualized by Cytoscape software (Figure 4B).

### 3.5. Immune Infiltration Analysis of Pathway Genes

Immune cells play an indispensable role in the treatment and development of tumors. Infiltration of immune cells can both remove tumor cells and play an anti-tumor role, thereby preventing disease progression; at the same time, tumor cells can escape the attack of immune cells by manipulating immune cells and thus promoting tumor progression. The CIBERSORT package was developed by Stanford University and is capable of accurately understanding the distribution of different types of immune cells in tissues. We applied the CIBERSORT package to calculate the infiltration of immune cells in each sample and then further analyzed the correlation between immune cells and between immune cells and pathway genes (Figure 5). A strong positive correlation between GSTP1 and activated mast cells; a strong positive correlation between activated mast cells and memory B cells, plasma cells, T follicular helper cells, activated NK cells, and resting mast cells and a strong negative correlation between activated mast cells and activated CD4 memory T cells, γδ T cells, M0 macrophages, and M2 macrophages can be observed in the figure.

### 3.6. Evaluate and Verify the Diagnostic Ability and Expression Level of Pathway Genes

We analyzed the diagnostic ability of pathway genes for prostate cancer by applying “pROC” to calculate AUC scores for receiver operating characteristic (ROC) curve analysis. To assess whether the pathway genes could be differentially expressed in the internal and external datasets, we applied R software to analyze the expression levels of the pathway genes between the control group and the experimental group in the dataset. Figure 6A–D are a visualization of the differential analysis of the diagnostic ability and expression levels of the pathway genes in the internal and external validation sets. The GSTP1 shown in the figure has better diagnostic ability in both training and validation sets with AUC values of 0.885 and 0.760, respectively. All seven pathway genes can be differentially expressed in both training and validation sets, and the expression trend of GSTP1 in different datasets is the same, i.e., the expression level is lower than that of normal tissues in the tumor tissues, which further proves that our selected GSTP1 gene has robust diagnostic ability. We used immunohistochemical results from the Human Protein Atlas (https://www.proteinatlas.org/ (accessed on 5 August 2024)) to show that GSTP1 is predominantly expressed in the cytoplasm and is expressed higher in normal tissues than in prostate cancer tissues (Figure 6E,F).

### 3.7. Constructing Prostate Cancer Prediction Models Based on Machine Learning and Pathway Genes

LASSO regression is generally used for feature selection. In this study, we used the “glmnet” package (v 4.1-8) to perform LASSO regression analysis on the training set to select pathway genes with a large number of features and used 10-fold cross-validation to train the model. Based on λ.min, 0.01310187 was identified as an appropriate λ value. Finally, three pathway genes (GSTP1, GSTM5, RRM2) with non-zero coefficients were obtained for constructing the prostate cancer prediction model (Appendix A). The formula of the prediction model is as follows: Riskscore = (−0.33351 × GSTP1 expression) + (−0.38436 × GSTM5 expression) + (1.182265 × RRM2). We performed ROC analysis of the model on the training set and validation set to evaluate the prediction accuracy of the model (Figure 7A–C). As shown in the figure, the constructed LASSO regression model has an AUC value of 0.949 in the training set and 0.789 in the validation set, indicating that the model has a robust diagnostic capability. Details of the LASSO scores for each sample in the training and validation sets can be found in Appendix A.

### 3.8. Single-Cell Analysis of Cell-Specific Expression Patterns of Pathway Genes in Prostate Cancer

To investigate the cell-specific expression of pathway genes in prostate cancer, RNA-seq analysis was first performed on selected samples. The data were first preprocessed, including quality control, normalization, and dimensionality reduction (Appendix A). Cells in the control and experimental groups were clustered into 25 different clusters by the UMAP algorithm, and the 25 clusters were categorized into seven different cell types in prostate cancer cells based on the gene expression of different clusters (Figure 8A and Appendix A) We further analyzed the cellular expression characteristics of the pathway genes and found that the pathway genes were significantly differentially expressed between the tumor and normal groups (Figure 8B). GSTP1 was most highly expressed, mainly in the epithelial cells (Figure 8C). Therefore, we specifically examined the characteristics between groups with high and low GSTP1 expression in epithelial cells. It was found that the proportion of the group with high expression of GSTP1 in epithelial cells in prostate cancer was lower than that in normal prostate tissue (Figure 8D); GO analysis showed that high and low GSTP1 expression groups in epithelial cells were clearly different in biological processes such as calcium-dependent protein binding, cadherin binding involved in cell–cell adhesion, and S100 protein binding. (Figure 8E); KEGG analysis showed that there were significant differences in tight junctions and estrogen signaling pathways between the groups with high and low GSTP1 expression in epithelial cells (Figure 8F). We performed four enrichment analyses, “AUCell”, “UCell”, “singscore”, and “ssgsea”, on the two epithelial cells, evaluated the results of the differential analysis by robust rank aggregation (RRA), and sorted out the sets of differential genes that showed similar levels of enrichment among the four methods. The upregulation of the P53 signaling pathway and apoptosis in epithelial cells with high expression of GSTP1 can be seen in Figure 8G.

## 4. Discussion

In this study, we investigated the causal relationship between enzymes in Glutathione metabolism and prostate cancer based on public databases; GSTP1 has the potential to serve as a biomarker for prostate cancer and may inhibit the development of prostate cancer through interactions between immune cells or upregulation of the P53 pathway and apoptosis.

Glutathione is synthesized from glutamic acid, cysteine, and glycine, and its use as an important antioxidant has shown more promise in tumor therapy [12]. The GST family is classified into α(A), κ(K), μ(M), ω(O), pi(P), σ(S), θ(T), and ζ(Z) according to amino acid sequence and substrate specificity; it is involved in key functions such as detoxification, cellular signaling, post-translational modification, resistance to chemotherapeutic agents and tumor progression [13,14]. DNA methylation, the most common epigenetic modification, plays a role in tumor cells; promoter hypermethylation of the GSTP1 gene is detected in prostate cancer, and this alteration is strictly limited to malignant cells, including prostate cancer and prostatic intraepithelial neoplasia [15]. The literature, therefore, suggests that hypermethylation of GST1 may be the best and most promising epigenetic marker, which is currently available for prostate cancer [15]. Vera Hauptstock et al. demonstrate that hypermethylation of the GSTP1 promoter in prostate cancer correlates with repressive histone patterns and that depsipeptide reverses DNA methylation and reduces repressive histone modifications for epigenetic therapy of prostate cancer [16]. Steven T. Okino et al. similarly concluded that the silencing of GSTP1 increases tumor susceptibility and promotes tumor progression and found that DNA methylation and histone modification silenced GSTP1 expression in prostate cancer; the order of alteration of these two epigenetic marks remains controversial [17]. Clare Stirzaker discovered that the accumulation of DNA methylation eventually leads to histone methylation, i.e., the sequence of DNA methylation precedes histone modification [18]. In contrast, Maria Strunnikova found that histone inactivation can trigger DNA methylation of the RASSF1A promoter, i.e., histone modification precedes DNA methylation [19]. Furthermore, Rajnee Kanwal et al. suggested that loss of GSTP1 expression due to methylation increases susceptibility to oxidative stress-induced DNA damage [20]. Regardless of the order, however, epigenetic alterations silence genes, which further play a role in tumors.

Immune infiltration analysis in this study showed a strong positive correlation between GSTP1 and activated mast cells, and mast cells had different degrees of positive and negative correlations with other immune cells. As an important member of the immune system, mast cells play an important role in anti-inflammation or pro-inflammation and are associated with the development of cancer [21]. In many studies, mast cells have been considered immune cells that play a protumorigenic role. A review on mast cells and colorectal cancer suggests that mast cells may promote colorectal cancer progression by secreting cytokines [22]. Enrique Zudaire et al. have shown that adrenomedullin modulates mast cells to promote tumor progression [23]. However, most epidemiological evidence supports a negative correlation between mast cells and tumor progression in breast, lung, and colon tumors, and Mark J Sinnamon demonstrated the protective role of mast cells in colon malignancies through animal experiments and stated that the anticancer effects of mast cells may be due to the different locations of mast cells [21]. GSTP1 may play an anticancer role in the prostate through activated mast cells. In single-cell analyses, different levels of epithelial GSTP1 expression differed significantly in different biological processes or metabolic pathways. For example, GO analysis showed that S100 protein binding was significantly different in high and low GSTP1 expression groups. It has been shown that S100 protein is associated with cell proliferation, metastasis, angiogenesis, and immune escape in cancer [24,25]. For instance, the expression of S100 A4 in hepatocellular carcinoma tissues correlates with progression [26]; the expression of S100 A8 and S100 A9 is regulated by hypoxia and HIF-1 in prostate cancer, and S100 A9 is associated with cancer recurrence [27]; real-time quantitative PCR reveals upregulated expression of S100 A2 and S100 P in metastatic non-small cell lung cancer and an overall survival benefit in patients with low S100 P expression [28]. KEGG analysis showed significant differences in the estrogen signaling pathway between groups with high and low GSTP1 expression. Prostate cancer is a highly androgen-dependent tumor, and inhibition of androgens has become an important tool in the treatment of prostate cancer [29,30]. Camille Lafront et al. found that the estrogen signaling pathway reprograms prostate cancer metabolism, which is linked to disease progression, and can be exploited for therapy [31]. All of these showed that GSTP1 expression is associated with the development and progression of prostate cancer. Single-cell enrichment analysis showed that P53 and apoptotic pathways were upregulated in the GSTP1 high-expression group. P53, as an important tumor suppressor, plays a central role in the cell cycle, and its inactivation is a hallmark of cancer [32,33,34]. It is regulated at multiple levels and, therefore, plays an important role in various aspects of the cell cycle, apoptosis, and anticancer [35]. An animal experiment showed that P53 restoration in mice led to the regression of lymphomas and sarcomas by inducing apoptosis and inhibiting cell growth [36], and several other in vivo studies have similarly reported the effects of restoring the function of P53 on tumors [37,38]. Apoptosis, as a form of programmed cell death, has been extensively studied in the field of tumor therapy due to its tumor-suppressive effects [39]. Apoptosis is categorized into the endogenous pathway mediated by B-cell lymphoma-2 (BCL-2) and the exogenous pathway mediated by death receptors (DR) [40,41]. It has been demonstrated that apoptosis induced by both endogenous and exogenous pathways occurs in prostate epithelial cells after androgen deprivation or denervation therapy, which is important in the suppression of prostate cancer [42]. Death receptors are members of the tumor necrosis factor receptor (TNFR) superfamily [43], including tumor necrosis factor receptor 1 (TNFR1) and TNF-related apoptosis-inducing ligand (TRAIL) receptors. TRAIL-R2 (DR5) has been reported in the literature to be downregulated in prostate cancer and significantly downregulated in high-grade tumors [44]. The DR5 agonist lexatumumab has been used in clinical trials with some efficacy [45]. This suggests that apoptosis has a non-negligible role in the treatment of cancer. Therefore, GSTP1 may inhibit tumorigenesis and progression by upregulating the P53 signaling pathway and apoptosis. This also reflects that Glutathione metabolism plays a role in prostate cancer.

There are some limitations to this study. First, the MR analysis was limited to a European population, so it remains unknown whether the conclusions are applicable to non-European populations; second, the sample size of the MR analysis was small, and the thresholds were relaxed, which leads to the possibility of false-positive results. Third, the small sample size of the data downloaded from the GEO database has some limitations, and the reliability of the results still needs to be further verified by in vivo and in vitro experiments. This study provides a genetic basis for the relationship between GSTP1 and prostate cancer risk and analyzes the molecular mechanism of GSTP1 in prostate cancer, but the relationship still needs to be further explored by experiments.

## 5. Conclusions

In summary, the molecular mechanism of GSTP1 in prostate cancer was revealed based on machine learning and single-cell analysis. We believe that GSTP1 is able to reduce the risk of prostate cancer, which is expected to be a potential therapeutic target for prostate cancer, and also allows us to recognize the non-negligible role of GSH metabolism for prostate cancer.

## Figures and Tables

**Figure 1 biomedicines-13-01051-f001:**
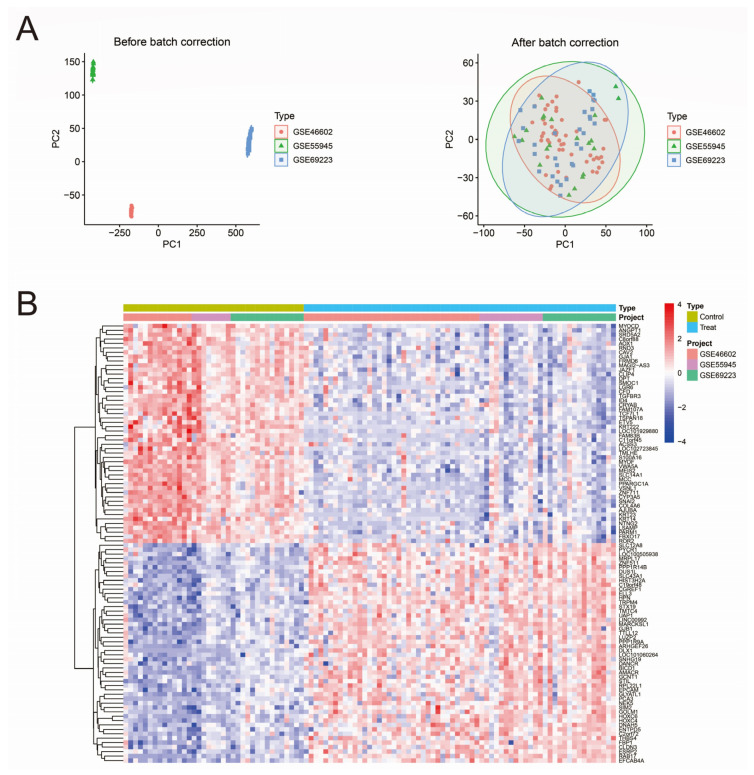
Data merging and differential gene analysis. (**A**) Visualization of PCA downscaling before and after data merging. (**B**) Heatmap of differentially expressed genes. (**C**) Volcano plot of differentially expressed genes. The red and blue dots represent upregulated and downregulated differentially expressed genes, respectively.

**Figure 2 biomedicines-13-01051-f002:**
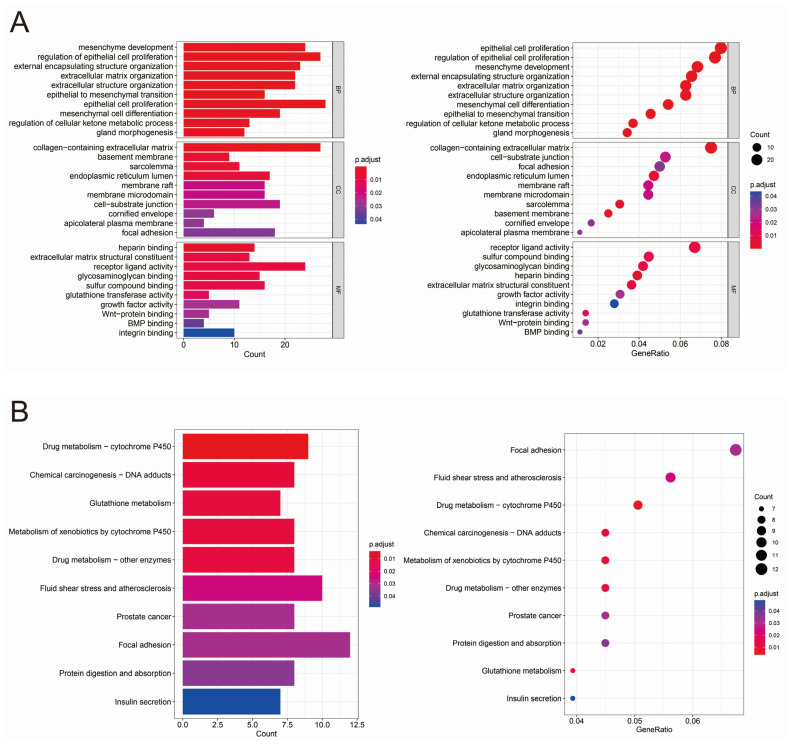
GO and KEGG analysis of differentially expressed genes. (**A**) GO analysis of differentially expressed genes. (**B**) KEGG analysis of differentially expressed genes.

**Figure 3 biomedicines-13-01051-f003:**
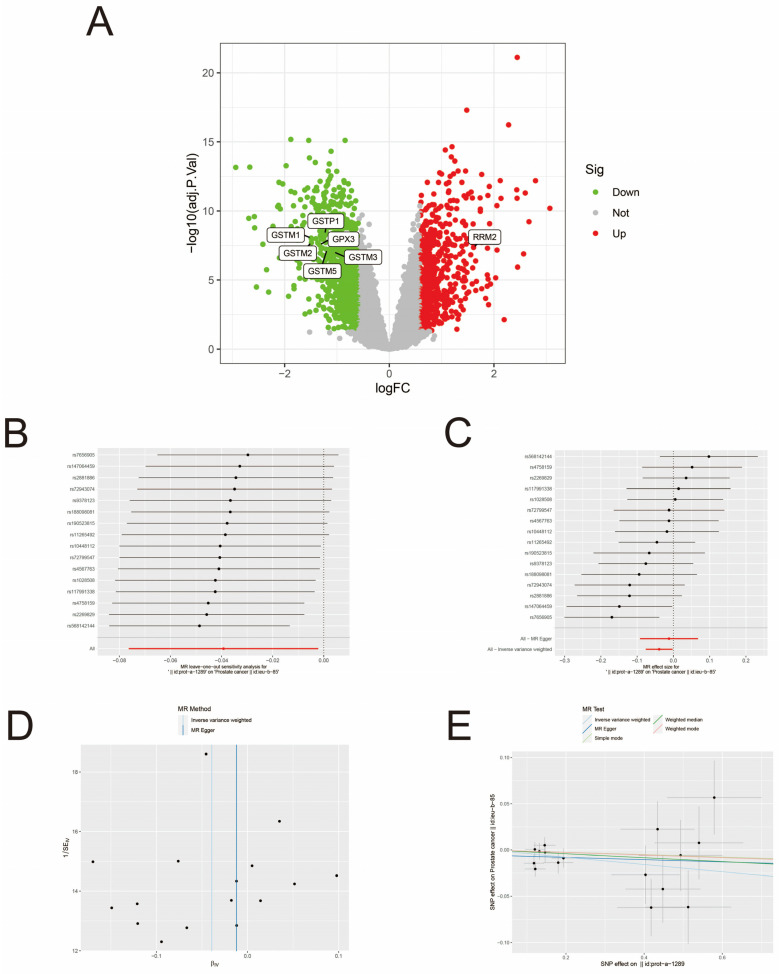
MR analysis of Glutathione metabolism and prostate cancer. (**A**) Volcano diagrams of pathway genes. (**B**) Leave-one-out analysis of the causal relationship between GSTP and prostate cancer. (**C**) Forest plot of the causal relationship between GSTP and prostate cancer. (**D**) Funnel plot of the causal relationship between GSTP and prostate cancer. (**E**) Scatter plot of the causal relationship between GSTP and prostate cancer. (**F**) Leave-one-out analysis of the causal relationship between GSOP1 and prostate cancer. (**G**) Forest plot of the causal relationship between GSOP1 and prostate cancer. (**H**) Funnel plot of the causal relationship between GSOP1 and prostate cancer. (**I**) Scatter plot of the causal relationship between GSOP1 and prostate cancer.

**Figure 4 biomedicines-13-01051-f004:**
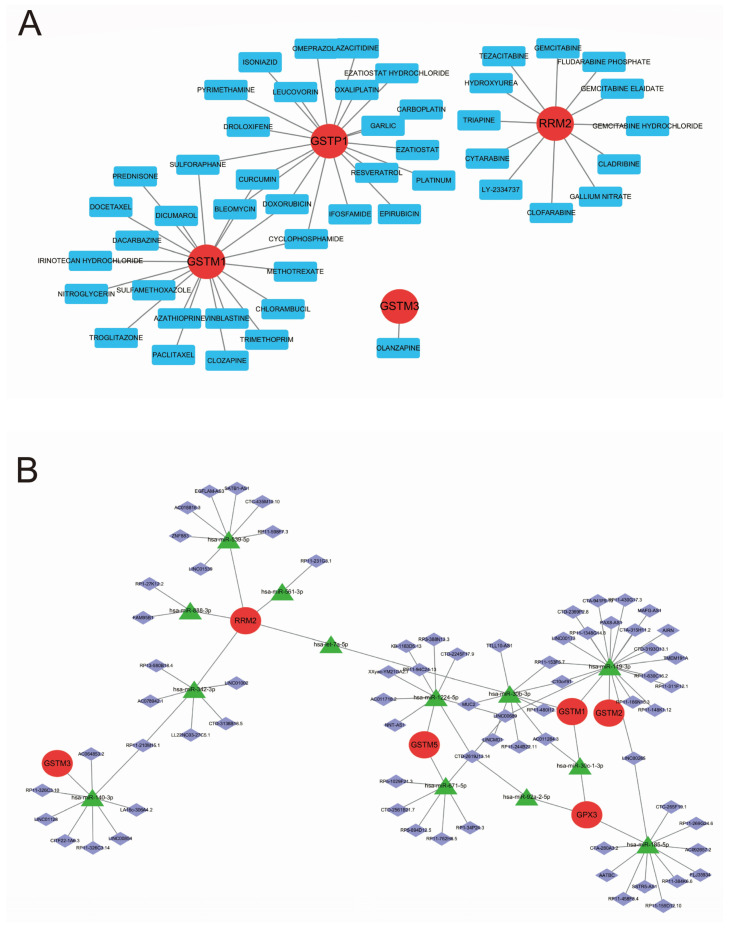
Construct gene–drug regulatory network and ceRNA regulatory network. (**A**) Gene–drug regulatory network. (**B**) ceRNA regulatory network.

**Figure 5 biomedicines-13-01051-f005:**
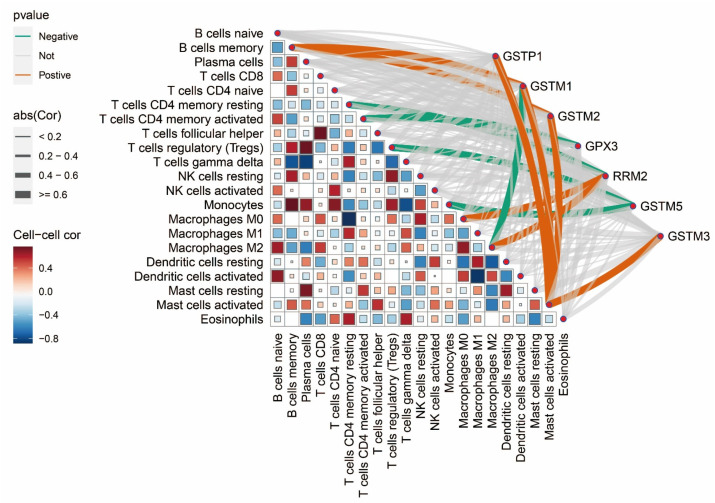
Immune infiltration analysis of pathway genes. The darker the color between immune cells, the stronger the correlation; red represents a positive correlation, while blue represents a negative correlation. The thicker the line between a gene and an immune cell, the greater the correlation; orange represents a positive correlation, while green represents a negative correlation.

**Figure 6 biomedicines-13-01051-f006:**
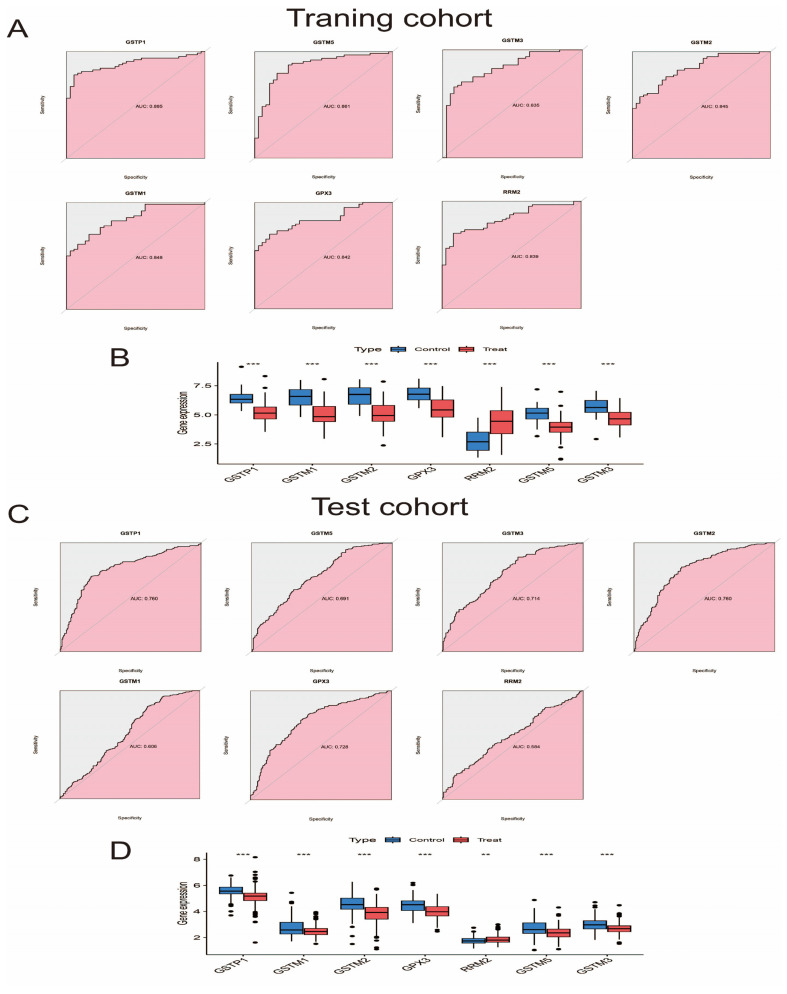
Differences in diagnostic ability and expression levels of pathway genes in the training and validation sets. (**A**) ROC curves for pathway genes in the training set. (**B**) Box plots of expression levels of pathway genes in training set. (**C**) ROC curves for pathway genes in the validation set. (**D**) Box plots of expression levels of pathway genes in validation set. (**E**) Immunohistochemical findings of GSTP1 in normal prostate tissue (scale bars: 200 μm). (**F**) Immunohistochemical results of GSTP1 in prostate cancer tissues (scale bars: 200 μm). **: *p* < 0.01; ***: *p* < 0.001.

**Figure 7 biomedicines-13-01051-f007:**
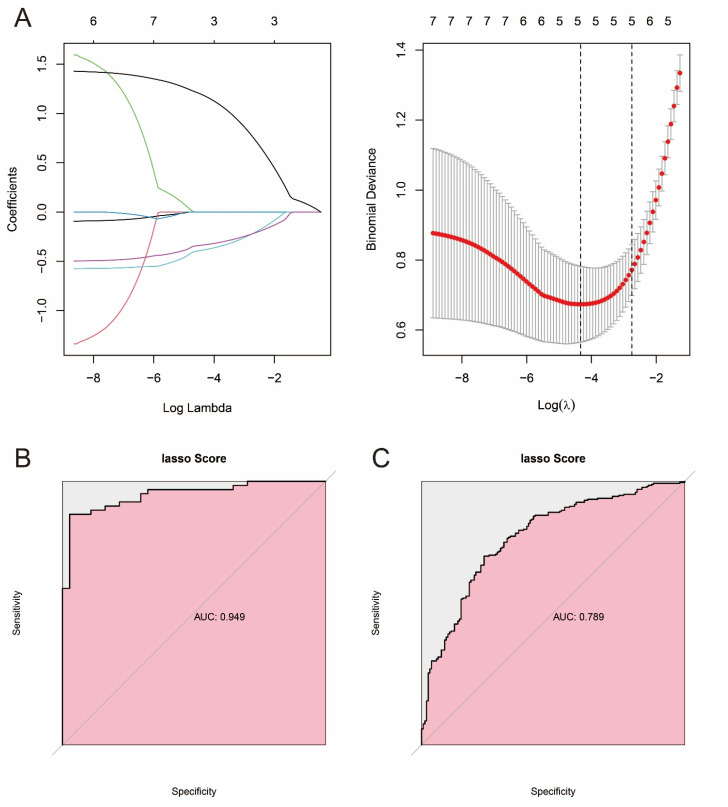
Construction and validation of LASSO regression model. (**A**) Screening hub genes from pathway genes by LASSO regression. (**B**) ROC curves for the LASSO regression model in the training set. (**C**) ROC curves for the LASSO regression model in the validation set.

**Figure 8 biomedicines-13-01051-f008:**
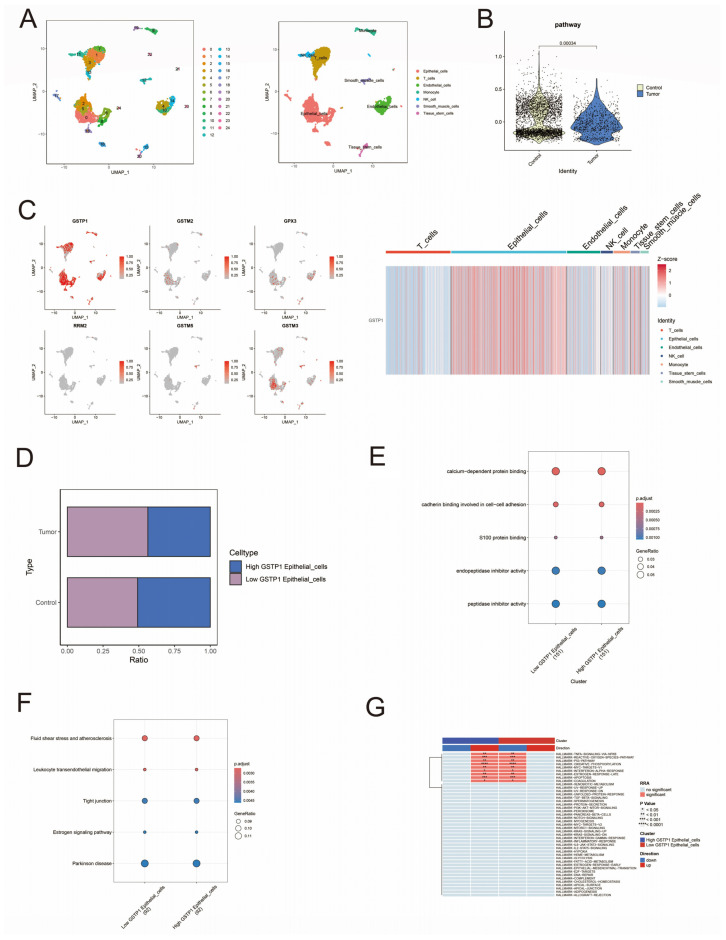
Single-cell analysis of pathway genes in prostate cancer. (**A**) UMAP plots in single-cell analysis. Different point colors represent different cell clusters or cell types. (**B**) Differences in expression levels of pathway genes in prostate cancer and normal controls. (**C**) Distribution of pathway genes in different cell types. (**D**) Proportion of subpopulations of high expressing and low expressing cells of GSTP1 in control and tumor groups. (**E**) GO analysis of high expressing and low expressing cell subpopulations of GSTP1. (**F**) KEGG analysis of high expressing and low expressing cell subpopulations of GSTP1. (**G**) Enrichment analysis of high expressing and low expressing cell subpopulations of GSTP1.

**Table 1 biomedicines-13-01051-t001:** Detailed information about the dataset.

GWAS Database			
Phenotype	Consortium	Sample Size	GWAS ID
**Exposure**			
Glutathione peroxidase 1 measurement	NA	10,708	ebi-a-GCST90019397
Glutathione peroxidase 7	NA	3301	prot-a-1265
Glutathione S-transferase A1	NA	3301	prot-a-1283
Glutathione S-transferase A3	NA	3301	prot-a-1284
Glutathione S-transferase A4	NA	3301	prot-a-1285
Glutathione S-transferase kappa 1	NA	3301	prot-a-1286
Glutathione S-transferase Mu 1	NA	3301	prot-a-1287
Glutathione S-transferase omega-1	NA	3301	prot-a-1288
Glutathione S-transferase P	NA	3301	prot-a-1289
Glutathione S-transferase theta-2B	NA	3301	prot-a-1291
Lactoylglutathione lyase	NA	3301	prot-a-1218
S-Formylglutathione hydrolase	NA	3301	prot-a-989
Cysteine-glutathione disulfide	NA	1997	met-a-680
Glutathione S-transferase Pi	NA	NA	prot-c-4911_49_2
**Outcome**			
Prostate cancer	PRACTICAL	140,254	ieu-b-85
**GEO Database**			
**Accession**	**Sample Source**	**Sequencing Type**	**Dataset Usage**
GSE46602	Prostate cancer	Array	Training dataset
GSE55945	Prostate cancer	Array	Training dataset
GSE69223	Prostate cancer	Array	Training dataset
GSE62872	Prostate cancer	Array	Testing dataset
GSE193337	Prostate cancer	RNA-seq	Single cell analysis

NA: No specifics are indicated in the database.

**Table 2 biomedicines-13-01051-t002:** Details of Drug-Gene Regulatory Network.

Gene	Drug	Interaction_Types	Sources
*GSTP1*	EZATIOSTAT HYDROCHLORIDE	inhibitor	ChemblInteractions
*GSTP1*	CARBOPLATIN		NCI|CIViC
*GSTP1*	CURCUMIN		NCI
*GSTP1*	LEUCOVORIN		PharmGKB
*GSTP1*	BLEOMYCIN		PharmGKB
*GSTP1*	ISONIAZID		PharmGKB
*GSTP1*	OXALIPLATIN		NCI|PharmGKB
*GSTP1*	AZACITIDINE		NCI
*GSTP1*	IFOSFAMIDE		NCI
*GSTP1*	GARLIC		NCI
*GSTP1*	EZATIOSTAT		TdgClinicalTrial|TTD
*GSTP1*	SULFORAPHANE		NCI
*GSTP1*	PYRIMETHAMINE		PharmGKB
*GSTP1*	PLATINUM		PharmGKB
*GSTP1*	CYCLOPHOSPHAMIDE		NCI|PharmGKB
*GSTP1*	DROLOXIFENE		NCI
*GSTP1*	RESVERATROL		NCI
*GSTP1*	OMEPRAZOLE		NCI
*GSTP1*	DOXORUBICIN		PharmGKB
*GSTP1*	EPIRUBICIN		PharmGKB
*GSTP1*	IRINOTECAN HYDROCHLORIDE		NCI
*GSTP1*	CYTARABINE		NCI
*GSTP1*	VITAMIN E		NCI
*GSTP1*	MELPHALAN		NCI
*GSTP1*	ALCOHOL		NCI
*GSTP1*	ADRIAMYCIN		NCI
*GSTP1*	PREDNISONE		NCI
*GSTP1*	EXATECAN MESYLATE		NCI
*GSTP1*	THIOTEPA		PharmGKB
*GSTP1*	CAMPTOTHECIN		NCI
*GSTP1*	DOCETAXEL		NCI
*GSTP1*	FLUOROURACIL		PharmGKB
*GSTP1*	DECITABINE		NCI
*GSTP1*	PACLITAXEL		CIViC
*GSTP1*	BUSULFAN		NCI|PharmGKB
*GSTP1*	DEXAMETHASONE		NCI
*GSTP1*	SELENOMETHIONINE		NCI
*GSTP1*	DITIOCARB		NCI
*GSTP1*	MISONIDAZOLE		NCI
*GSTP1*	LYCOPENE		NCI
*GSTP1*	DAUNORUBICIN		PharmGKB
*GSTP1*	RIFAMPIN		PharmGKB
*GSTP1*	ETOPOSIDE		PharmGKB
*GSTP1*	CANFOSFAMIDE		TdgClinicalTrial|TTD
*GSTP1*	CISPLATIN		NCI|CIViC|PharmGKB
*GSTP1*	HYDROQUINONE		NCI
*GSTP1*	PERFOSFAMIDE		NCI
*GSTP1*	VERAPAMIL		NCI
*GSTM1*	CYCLOPHOSPHAMIDE		PharmGKB
*GSTM1*	PACLITAXEL		PharmGKB
*GSTM1*	CURCUMIN		NCI
*GSTM1*	DOXORUBICIN		PharmGKB
*GSTM1*	DICUMAROL		NCI
*GSTM1*	VINBLASTINE		PharmGKB
*GSTM1*	SULFORAPHANE		NCI
*GSTM1*	CHLORAMBUCIL		NCI
*GSTM1*	CLOZAPINE		PharmGKB
*GSTM1*	AZATHIOPRINE		PharmGKB
*GSTM1*	TROGLITAZONE		NCI
*GSTM1*	NITROGLYCERIN		NCI
*GSTM1*	IRINOTECAN HYDROCHLORIDE		NCI
*GSTM1*	METHOTREXATE		NCI
*GSTM1*	BLEOMYCIN		PharmGKB
*GSTM1*	PREDNISONE		NCI
*GSTM1*	SULFAMETHOXAZOLE		PharmGKB
*GSTM1*	DOCETAXEL		NCI
*GSTM1*	DACARBAZINE		PharmGKB
*GSTM1*	TRIMETHOPRIM		PharmGKB
*GSTM1*	BUSULFAN		PharmGKB
*RRM2*	HYDROXYUREA	inhibitor	ChemblInteractions|TTD
*RRM2*	FLUDARABINE PHOSPHATE	inhibitor	ChemblInteractions
*RRM2*	GALLIUM NITRATE	inhibitor	ChemblInteractions
*RRM2*	CLOFARABINE	inhibitor	ChemblInteractions
*RRM2*	GEMCITABINE HYDROCHLORIDE	inhibitor	ChemblInteractions
*RRM2*	GEMCITABINE	inhibitor	ClearityFoundationClinicalTrial|TTD
*RRM2*	TEZACITABINE	inhibitor	ChemblInteractions
*RRM2*	CLADRIBINE	inhibitor	PharmGKB
*RRM2*	LY-2334737		TTD
*RRM2*	CYTARABINE		PharmGKB
*RRM2*	TRIAPINE		TdgClinicalTrial|TTD
*RRM2*	GEMCITABINE ELAIDATE		TTD
*GSTM3*	OLANZAPINE		PharmGKB

## Data Availability

No raw data were generated for this study. The sources of data used are cited in the text. All other relevant data are included in the main text.

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
