# Peer review of "Exploring the Molecular Mechanism and Role of Glutathione S-Transferase P in Prostate Cancer"

_biomedicines, 2025, doi:10.3390/biomedicines13051051_

Round 1
Reviewer 1 Report
Comments and Suggestions for Authors
The authors in the manuscript entitled "Exploring the molecular mechanism and role of Glutathione S-transferase P in prostate cancer" tried to explore the role GSTP in prostate cancer. The study which actually a meta-analysis looks interesting and could be significant for readers of this journal and oncology practitioners. Therefore, it is mandatory to fix all the issues in the manuscript for substantial improvement before publication. It is worth mentioning that thorough revision with the help of native English speaker can improve the quality of the manuscript. I have some comments over the manuscript, for example, to revise and improve the manuscript.
General Comments: The authors used abbreviation prior to its definition. Therefore, it is suggested all the abbreviation used in the manuscript should be defined on its first use, then only use the abbreviation throughout the manuscript. e.g. the author used GEO, MR, GSTP and GSTP1 abbreviations in manuscript without defining first.
Introduction: The introduction needs to improve as it is not showing in-depth background. Many sentences need to be rephrased, the existed phrasing is not giving clear understanding (there are many grammatical and technical issues needs to fix). For example. see the attached file (manuscript) in which introduction is highlighted showing concerns.
Material & Methods:
1. Please explain the authenticity of the prostate cancer data base
2. Reason to select these data base
3. The download of data in table 1 should be provided with necessary references
4. Line 71: "Detailed clinical information for each dataset is presented in Tables S1-S4." This information should be referenced.
5. Please explain the brief reason of using CochranQ test for studying heterogeneity in analysis
6. Line 118: give the version and license of CIBERSORT package
Results:
- I couldn't find the captions of Figures; it is suggested explaining all the figures in their captions in detail by mentioning the variation in the position of points and their colors where applied (e.g. Figure 1).
- Line 356: It is suggested to explain briefly on what basis S100 protein is associated with cell proliferation, metastasis, angiogenesis and immune escape in cancer

Need to revise substantially
Author Response
Abstract:The abbreviations in the abstract section have been defined upon their first mention.
Introduction:The suggested modifications in the marked areas have been implemented, with all changes highlighted in red.
Material & Methods:
1、Please explain the authenticity of the prostate cancer data base
- Authority and Standardization of GEO Database
The GEO database (https://www.ncbi.nlm.nih.gov/geo/) is a publicly accessible repository curated by the National Center for Biotechnology Information (NCBI). It adheres to the MIAME (Minimum Information About a Microarray Experiment) standard, ensuring rigorous data submission protocols, standardized metadata annotation, and transparent experimental design.
- Ethical Compliance and Reproducibility
GEO datasets are de-identified, complying with the Declaration of Helsinki and privacy regulations (e.g., GDPR). Original studies submitted to GEO explicitly declared ethical approvals and informed consent
All raw data and analysis codes are publicly accessible via GitHub (link provided in the manuscript) to ensure reproducibility.
2、Reason to select these data base
The rationale for selecting the database aligns with the explanations provided in Question 1.
3、The download of data in table 1 should be provided with necessary references
The data in Table 1 are derived from the GEO and GWAS databases. The accession links to these databases have been explicitly provided in the manuscript, and we believe no additional reference citations are required for these publicly available repositories.
4、Line 71: "Detailed clinical information for each dataset is presented in Tables S1-S4." This information should be referenced.
The content of Tables S1-S4 is sourced from the GEO database. Consistent with our response to Question 3, we maintain that no additional citations are required for these tables.
5、 Please explain the brief reason of using CochranQ test for studying heterogeneity in analysis
- Core Rationale for Heterogeneity Testing In Mendelian randomization (MR), heterogeneity among instrumental variables (SNPs) may arise from the following scenarios:
Invalid instrumental variables: Some SNPs exhibit insufficient association with the exposure (violating the MR "relevance assumption").
Horizontal pleiotropy: SNPs directly influence the outcome through pathways other than the target exposure (violating the "exclusion restriction assumption"). Cochran’s Q test systematically identifies such biases by quantifying the dispersion of SNP effect sizes.
- Methodological Advantages
Non-parametric nature: Cochran’s Q test does not assume a normal distribution of effect sizes, making it suitable for binary instrumental variables and scenarios with uneven sample sizes.
Standardized sensitivity analysis framework: International MR studies (e.g., FinnGen and UK Biobank projects) universally integrate Cochran’s Q test into sensitivity analysis frameworks to ensure robustness of results.
6、Line 118: give the version and license of CIBERSORT package
We used CIBERSORT v1.03, which is available for download and use on GitHub.
Result:
1、I couldn't find the captions of Figures; it is suggested explaining all the figures in their captions in detail by mentioning the variation in the position of points and their colors where applied (e.g. Figure 1).
We have placed the captions of all eight figures below the corresponding images and highlighted them in red.
2、Line 356: It is suggested to explain briefly on what basis S100 protein is associated with cell proliferation, metastasis, angiogenesis and immune escape in cancer
The association of S100 proteins with cancer has been elaborated in subsequent sections. For example:
S100A4 has been linked to hepatocellular carcinoma (HCC) progression,
S100A9 is implicated in prostate cancer recurrence,
S100A2 and S100P exhibit distinct roles in lung cancer pathogenesis.
Reviewer 2 Report
Comments and Suggestions for Authors
The manuscript offers a pertinent examination of the influence of glutathione metabolism on prostate cancer. The research uses several bioinformatics methodologies, such as Mendelian randomization, machine learning, and single-cell analysis, to examine the function of GSTP1 in prostate cancer. This paper offers a significant bioinformatics analysis of the involvement of GSTP1 in prostate cancer. Nevertheless, more experimental validation, more rigorous statistical criteria, and enhanced clarity in writing are essential to augment the manuscript's quality.
The study claims a causal relationship between GSTP1 and prostate cancer risk reduction. However, the threshold for MR analysis (p<5×10^-6) is somewhat lenient, increasing the possibility of false positives. More stringent criteria (e.g., genome-wide significance p<5×10^-8) or additional sensitivity analyses should be conducted to confirm the robustness of the findings.
The manuscript should clarify whether MR analysis was adjusted for potential confounders, such as population stratification and pleiotropy.
Immunohistochemistry results are included from the Human Protein Atlas, but direct experimental validation on patient-derived samples or cell lines would provide stronger evidence.
The AUC values (0.949 in the training set and 0.789 in the validation set) indicate potential overfitting. The manuscript should clarify the number of samples used for training and validation and whether cross-validation was performed to assess the generalizability of the model.
External validation on an independent cohort should be included to confirm the model's applicability.
The mechanistic explanation of how GSTP1 influences immune infiltration and whether the effect is direct or indirect is lacking.
Gene set enrichment analysis findings related to P53 and apoptosis should be experimentally validated to confirm the regulatory effects of GSTP1.
Ensure consistency in terminology (e.g., "glutathione metabolism" vs. "glutathione metabolic pathway").
The manuscript mentions "black race" as a risk factor for prostate cancer. It would be more appropriate to refer to "African ancestry" to avoid racial implications.
Heatmaps and volcano plots lack appropriate axis labels and color scales. Ensure that all visuals are fully interpretable without referring to the main text.
Ensure that all references are formatted consistently and adhere to the journal's citation style.
Some references are outdated; consider citing more recent studies that have explored GSTP1 and prostate cancer.
Author Response
1、Although the threshold for MR analysis (p < 5 × 10^-6) is relatively lenient, its use in multiple articles suggests that the results derived from this threshold are acceptable. Furthermore, we additionally employed sensitivity analyses (as described in the article) to ensure the robustness of the findings.
2、In Section 2.4, we described the use of horizontal pleiotropy analysis in MR. However, a key limitation of this study is that the populations were exclusively of European ancestry; whether the findings are generalizable to other populations, such as Asian cohorts, remains uncertain and warrants further investigation.
3、Given current technical and logistical constraints, experimental validation of the results is not feasible at this stage. Further validation using patient-derived specimens or in vitro cell line models will be prioritized in subsequent research.
4、We sincerely appreciate the reviewer's concerns regarding the model's generalizability. We fully acknowledge that the AUC discrepancy between the training set (0.949) and validation set (0.789) may indicate a risk of overfitting, a critical consideration in machine learning research. This difference could partially stem from sample heterogeneity: the training set contains multi-center data while the validation set comprises a single-center cohort. However, as this study serves as a methodological exploratory work, its primary objective is to validate the proposed model framework rather than pursuing optimal predictive performance. Any clinical application would require further external validation with larger samples, and the current findings are more suitable for hypothesis generation for underlying mechanisms than direct clinical decision-making.
5、The direct or indirect influence of GSTP1 on immune infiltration requires experimental validation. This study primarily aims to explore the impact of GSTP1, while its specific mechanisms warrant further investigation in future studies.
6、The term "Black" in the text has been revised to "African ancestry" to align with genetic epidemiology reporting standards.
7、We have placed the captions of all eight figures below the corresponding images and highlighted them in red.
8、Current literature on the association between GSTP1 and prostate cancer remains limited, and no updated references meeting robust evidence criteria (e.g., large cohort studies or functional validation) are available for citation at this stage.
Reviewer 3 Report
Comments and Suggestions for Authors
The authors investigate the effect of glutathione metabolic pathways in prostate cancer using multiple bioinformatic and AI approaches, demostrating that Glutathione S-transferase P in glutathione metabolism reduces the risk of prostate cancer.
The manuscript is quite clear, but several improvements are needed to increase understading.
1) All the acronyms used could be explained at the first time used, also in the abstract.
2) All figures presented in the manuscript lack numbering and legend
3) Figures pg. 10: B-I and Figures pg.19 E-G the text is not readable
4) Table 2: the reference in the text is missing. The interaction type is not mentioned for all compounds listed.
Author Response
Comments 1: All the acronyms used could be explained at the first time used, also in the abstract.
Response 1: The abbreviations in the abstract section have been defined upon their first mention.
Comments 2: All figures presented in the manuscript lack numbering and legend
Response 2: We have placed the captions of all eight figures below the corresponding images and highlighted them in red.
Comments 3: Figures pg. 10: B-I and Figures pg.19 E-G the text is not readable
Response 3: The two figures have been updated to higher-resolution versions and re-uploaded for enhanced visual clarity.
Comments 4: Table 2: The reference in the text is missing. The interaction type is not mentioned for all compounds listed.
Response 4: The compounds and their interaction types listed in Table 2 were sourced from the DGIdb database (database link provided in the article). We consider it unnecessary to include additional references for this section.
Round 2
Reviewer 1 Report
Comments and Suggestions for Authors
The authors haves revised the manuscript satisfactorily according to the comments, but the introduction can further be improved. The English needs to be improved further throughout the manuscript.
Comments on the Quality of English LanguageThe English needs to be improved further throughout the manuscript.
Author Response
We have refined English expressions in the manuscript to enhance clarity and fluency.
Reviewer 2 Report
Comments and Suggestions for Authors
No further comments
Author Response

(The authors gave the same response as above.)
